# Flavin Adenine Dinucleotide (FAD) and Pyridoxal 5′-Phosphate (PLP) Bind to Sox9 and Alter the Expression of Key Pancreatic Progenitor Transcription Factors

**DOI:** 10.3390/ijms232214051

**Published:** 2022-11-14

**Authors:** Zeyaul Islam, Noura Aldous, Sunkyu Choi, Frank Schmidt, Borbala Mifsud, Essam M. Abdelalim, Prasanna R. Kolatkar

**Affiliations:** 1Diabetes Research Center (DRC), Qatar Biomedical Research Institute (QBRI), Hamad Bin Khalifa University (HBKU), Qatar Foundation, Doha P.O. Box 34110, Qatar; 2College of Health and Life Sciences, Hamad Bin Khalifa University (HBKU), Qatar Foundation, Doha P.O. Box 34110, Qatar; 3Proteomics Core, Weill Cornell Medicine, Doha P.O. Box 24144, Qatar

**Keywords:** pancreatic progenitors, Sox9, FAD, PLP, transcription factor, Sox family

## Abstract

Cofactor flavin adenine dinucleotide (FAD), a compound with flavin moiety and a derivative of riboflavin (vitamin B_2_), is shown to bind to Sox9 (a key transcription factor in early pancreatic development) and, subsequently, induce a large increase in markers of pancreatic development, including Ngn3 and PTF1a. Pyridoxal 5′-phosphate (PLP), the active form of vitamin B_6_, also binds to Sox9 and results in a similar increase in pancreatic development markers. Sox9 is known to be specifically important for pancreatic progenitors. Previously, there was no known link between FAD, PLP, or other co-factors and Sox9 for function. Thus, our findings show the mechanism by which FAD and PLP interact with Sox9 and result in the altered expression of pancreatic progenitor transcription factors involved in the pancreas development.

## 1. Introduction

Induced pluripotent stem cell (iPSC) development is a key process being studied widely due to the potential for modulating these cellular pathways towards specific differentiation outcomes such as cardiac or pancreatic tissues [1] for regenerative medicine. Although a multitude of proteins play major roles in these processes, a critical role is played by transcription factors. Transcription factors (TFs) comprise a large percentage of the total genes, and their interplay during development is an important step towards differentiation.

Among the TFs, the family of Sox proteins plays significant roles in diverse pluripotency and differentiation programs despite having highly conserved sequences within this family. Several studies have shown how single residue changes within the TF or single base changes in the respective genomic binding site can subtly alter the mutual recognition and result in completely changed differentiation outcomes [2]. Sox2 and Sox17 can adopt each other’s programs through single site changes within the HMG domain (DNA-binding) or the cognate cis element [3,4,5]. 

Combinatorial TF interactions drive the majority of transcriptional pathways at promoters and also at distal enhancer regions. Little is known of potential small-molecule metabolic factors which could potentially affect TF pathways. Nuclear receptor HNF4α has been shown to be modulated by endogenous linoleic acid using mass spectrometry to show direct binding, and could play a role in hepatic lipid metabolism [6,7]. MocR-like bacterial transcription factors (MocR-TFs) comprise N-terminal DNA-binding domains and a pyridoxal 5′-phosphate (PLP) binding site at the C-terminal portion. The capability to recognize specific DNA sequences and concurrently bind PLP allow these TFs to regulate the transcription of different metabolic processes involving vitamin B_6_ and amino acids [8,9].

The inhibitor of DNA binding and differentiation 2 (ID2) was also reported to be functionalized through a divalent ion [10]. However, these are unique and rare endogenous small-molecule metabolic factors which are known to interact with a TF and modulate the TF pathway. 

Sox9, a member of the SoxE family, is known to be important in pancreatic development [11]. A key role of Sox9 is to maintain pancreatic progenitor cells, but it also has additional roles in endocrine differentiation [12,13]. Sox9 also plays a role in hepatic and duodenal programs [14]. Sox9 perturbation is involved in campomelic dysplasia (CD), as Sox9 also has a key role in chondrogenesis. It is also noted that pancreatic dysfunction is often correlated with individuals suffering from CD [12].

Sox9 is composed of an N-terminal dimerization domain followed by an HMG domain (Figure 1a) and an extended disordered region at the C-terminal. The dimerization of Sox9 was observed to be important for its biological function, and alanine to glutamic acid at position 76 (A76E) mutation was shown to disrupt dimerization and its function [15,16,17]. Dimerization could also be installed via the interaction of the HMG domain and the dimerization domain of Sox9 [17,18]. Sox9 is shown to predominantly bind at promoters during foregut development [19]. Dimerization is also observed in Sox9 binding to Ngn3 promoter in pancreatic progenitor cells [20]. The abrogation of DNA-dependent dimerization of Sox9 is also associated with CD [15], as the loss of binding to enhancer elements leads to changes in downstream targets. In addition, Sox9 dimerization is also involved in heart valve development [21]. 

Recent analyses of β-cell neogenesis has been applied to create in vitro methods for producing glucose responsive β-cells [22]. The adult pancreas also exhibits β-cell neogenesis under certain conditions, such as partial duct ligation or cerulein treatment [23,24]. Thus, a better understanding of the mechanisms involved in β-cell production could lead to improved treatment either through better in vitro production of β-cells or in situ within the adult pancreas.

We present here two unique cofactors (Figure 1b), flavin adenine dinucleotide (FAD) and pyridoxal 5′ phosphate (PLP), which interact with Sox9 and alter pancreatic progenitor development. Using a combination of data from mass spectrometry of purified Sox9, as well as absorbance spectroscopy and several mutations of FAD/PLP binding residues, we show the mechanism of Sox9–FAD binding. Next, we introduce FAD and PLP in hiPSCs to show that there is a significant increase in markers of pancreatic progenitors. This is the first report of endogenous small-molecule metabolic cofactors which can modulate transcription in pancreatic development.

## 2. Results

Sox9HMG-sox_N domain (residues 1–173) was produced using bacterial expression, and the purified protein (Figure 2a–c) exhibited a yellowish-brown color (Figure 2d). A similar yellowish color was observed for the Sox9 HMG domain, but not observed with the Sox2 HMG domain protein (Appendix A) obtained in a similar manner. The yields from expression were similar for all forms of HMG domains. All the proteins also had high purity, and the molecular sizes of these highly conserved HMG domains (Sox9 (Figure 2a–c) and Sox2) were similar (~15 kD). 

Further analysis employing absorbance spectroscopy showed a peak at approximately 420 nm, characteristic of an FAD/PLP-bound protein [25,26], for all Sox9 constructs. This peak was pronounced and significant relative to the spectrum. The profile for the Sox2 HMG domain showed no obvious peak (Appendix A). 

The purified Sox9HMG-sox_N domain was subjected to 1D-PAGE in-gel digest and LC-MS/MS to identify the putative cofactor giving rise to the yellow color of the protein. Several peaks were detected suggestive of FAD, FMN, and PLP binding due to the changed intensity profile (Figure 3a,b) for specific peptide fragments. Six peptide sequences corresponding to either FAD or PLP binding with high probability were identified with six specific residues (K122, K141, K166, K167, H104, H165) potentially responsible for binding the cofactor. These specific residues were chosen because they were all within the HMG domain and all had both mass and MS/MS confirmation. Histidine residues are known to bind FAD [27,28], whereas lysine residues are known to bind PLP [29,30]. 

Sequence comparisons were made between Sox9 and several other Sox family members (Figure 4). The homology/identity of the HMG domains is high (>90%). H104 and K166 are primarily conserved within SoxE family members (Sox8,9,10) and not observed with most other Sox family members. The A76 residue, associated with dimerization, is also found only in SoxE family members. In addition, A76E, H165Q, and H165Y mutants are found in CD. A76S, K141E, K141N, K166N, K166E, K167T, H165R, K141Qfs*111 insertion-frameshift and K167 deletion were also found as somatic mutations in cancer patients, especially in colon adenocarcinoma.

Site-directed mutants of the six residues (individually converted to alanine), as well as the previously reported [15,17] dimerization mutant (A76E) of the Sox9HMG-sox_N domain, were purified and characterized by absorbance spectroscopy. All mutants showed an absorbance peak similar to the wild-type Sox9HMG-sox_N and HMG domain, except for H104A, H165A, and A76E, where the peak was completely absent (Figure 5a,b). 

The structural analysis of available Sox9 HMG domain structures (complexed with cognate DNA) showed that both the mouse (4S2Q) and human (4EUW) Sox9 structures are highly similar (Cα rmsd of ~ 1Å over 72 aligned residues.rmsd). The inspection of the dimerization domain and residues H104 and H165 showed that the residues could participate in a dimerization putatively, as they all lie on the same side of Sox9 (Figure 6).

hiPSC-derived pancreatic progenitors were generated in vitro and were initially treated with different FAD and PLP concentrations (Figure 7A) during the pancreatic progenitor stage (stage 4). The efficiency of pancreatic progenitor differentiation was determined by examining the expression of PDX1 and NKX6.1. Our immunostaining results showed that most of the pancreatic progenitor cells co-expressed PDX1 and NKX6.1 (Figure 7B), indicating the high differentiation efficiency. Pancreatic progenitors treated with FAD during the first 24 h showed increased expression levels of PTF1a, involved in the proliferation of multipotent progenitor cells (MPCs) and also a key pancreatic exocrine marker, showing an ~8 fold increase with 1 µM FAD treatment. NGN3, which is a pancreatic endocrine progenitor marker, also increased ~14 fold as a result of FAD treatment (Figure 8A). A similar pattern was observed in hiPSC-derived pancreatic progenitors treated with PLP, where PTF1a and NGN3 expression levels were increased in a dose-dependent manner (Figure 8B). hiPSC-derived pancreatic progenitors were also treated with FAD during the four days of the pancreatic progenitor stage of pancreatic differentiation. Western blotting showed increased levels of the transcription factors SOX9 and PTF1a upon FAD treatment (Figure 8C,D). SOX9 increased gradually with increased FAD concentrations, whereas PTF1a was highly increased in 0.5 µM and 1 µM FAD-treated pancreatic progenitors (Figure 8C,D).

Structure-based docking/in silico interaction was performed using the AutoDock Vina. The docking simulation was run at an exhaustiveness of 8 and was set to only output the lowest energy pose. The docking results of FAD and PLP with the HMG domain of Sox9 protein showed a significant binding affinity. FAD shows higher docking scores compared to PLP (for FAD, binding affinity of −7.4 Kcal/mol, and for PLP, −4.9 Kcal/mol). FAD docks at the interface of helices and was stabilized by several interactions, including H165 (Figure 9a). Similarly, PLP interacts in small grove and interacts with several residues, including K141 (Figure 9a).

## 3. Discussion

The critical role for TFs in stem cell biology and pancreatic development has been well established. The combinatorial complexity of TFs enabled via multiple bound enhancers and promoters was shown in pancreatic development [31]. Although the role of TFs has been investigated by many laboratories, there is little known about ligands and other co-factors which could influence the binding and subsequent functionality. Linoleic acid was previously observed to modulate nuclear receptor HNF4a using mass spectrometry to validate the binding [6,7]. The role of some bacterial cofactors, such as MocR-TFs [8,9], has also been shown previously, but little is known for human TFs. 

We report the role of cofactors FAD and PLP, which bind Sox9 and then help to modulate the subsequent binding to its cis element and subsequently affect the function at the cellular level. During the purification of Sox9 constructs (Sox9HMG-sox_N and HMG domain) from bacterial expression, a yellowish-brown color was seen in the purified protein (Figure 2d). Purified Sox2 HMG protein was clear, as seen previously with other Sox HMG domains, such as Sox17 and Sox4. The absorbance spectroscopy of Sox9 constructs also showed a peak at ~420 nm, corresponding to potential FAD/PLP cofactors, whereas the Sox2 HMG domain showed no peak (Figure 2d). 

MALDI-TOF was subsequently applied to further confirm the putative bound cofactors. The mass spectroscopic analysis showed that there were six significant peaks, corresponding to FAD, FMN, and PLP binding. Mutant proteins were made at the putative site of cofactor binding for all six peptides identified by 1D-PAGE in-gel digest and LC-MS/MS (K122, K141, K166, K167, H104, H165). The six mutant proteins, as well as a dimerization mutant (A76E), were subjected to absorbance spectroscopy. The dimerization of Sox9 is known to be important for function for several functions, such as chondrogenesis [16]. The characteristic peak expected at ~420 nm for all the putative cofactors was lost only in the mutants associated with the binding of FAD (H104, H165), as well as the dimerization mutant (Figure 5a,b). It is possible that K166 and K167 are next to each other and could possibly serve overlapping roles to bind PLP.

Sequence analysis showed that Sox9 residues K166 and A76 were the rare residues not conserved when compared to all the other Sox family members (Figure 4). This is a highly conserved family with >90% identity in the HMG domain. These non-conserved residues could explain the reason that FAD binds specifically to the SoxE family and no other Sox members. In addition, H104 is also part of a KxxPx’xxxP motif [32], where x’ is H104. This motif is also observed only in the SoxE family. These motifs are associated with promoting the binding/partnering of other proteins where the proline bracket allows the display of the intervening residues. Furthermore, H104 and K166 all lie on the same side of the HMG domain and likely are on the same side as A76, involved in dimerization. Thus, these residues are poised to affect dimerization and subsequent DNA binding and also would be affected when bound with cofactors which could aid in this dimerization and DNA-binding. Interestingly, H165Q, H165Y, and A76E mutants were found to be associated with CD. Additionally, A76S, K141E, K141N, K166N, K166E, K167T, H165R, K141Qfs*111 insertion-frameshift and K167 deletion were also found as somatic mutations in cancer patients, especially in colon adenocarcinoma. Additionally, an increased availability of both FAD (Folate) [33] and PLP (Vitamin B_6_) [34] has been observed to have an inverse relationship to diabetes occurrence in epidemiological studies. Thus, there is a likely role for the cofactor binding and subsequent dimerization to drive normal cell development, and the differentiation and perturbation of these residues disrupts pancreatic and likely chondrogenic programs, as well as leading to carcinoma.

hiPSCs subjected to pancreatic differentiation with the addition of FAD or PLP at the pancreatic progenitor stage were used to study the effects at cellular levels. The increase in SOX9 expression upon low concentrations of FAD treatment may have promoted pancreatic progenitor development, as SOX9 has been reported to be involved in PDX1 expression [35]. However, higher FAD concentrations, 0.5 µM and 1 µM, resulted in increased levels of SOX9 and PTF1a, indicated by Western blot. Pancreatic populations expressing high levels of SOX9 result in the formation of ductal cells in later stages, whereas high levels of PTF1a are responsible for promoting MPCs [36] in earlier stages, and later lead to acinar cells [35]. Hence, FAD/PLP treatment may result in controlling the balance between MPC proliferation or promoting exocrine development in later stages. In addition, the observed increase in NGN3 expression levels at the pancreatic progenitor stage suggests the potential for endocrine development. Taken together, these findings suggest that FAD and PLP treatment at the pancreatic progenitor stage is potentially useful for enhancing the generation of functional pancreatic beta cells.

The functional effects seen at the cellular level are consistent with the biochemical observations showing that FAD cofactor binding to Sox9 triggers an increase in the pancreatic progenitor program. Decreased PLP has been observed to be correlated with an increased rate of diabetes due to potential alterations in enzymes related to disturbed insulin activity [37]. Rat islet β-cells show the modulation of FAD in response to glucose [38] and the presence of FAD in human pancreatic β-cells [39]. The results presented in this paper show that cofactors such as FAD and PLP could potentially be affecting beta cell regeneration by affecting the developmental pathways via Sox9 to increase the pancreatic progenitor population. The biochemical results are well supported by the cell biology showing that the cofactor binding is not an adventitious artefact. The cofactors could be acting via the promotion of Sox9 dimerization facilitating Sox9 binding to its genomic location (Figure 9b). These metabolites which are present in the human pancreas could be sensors of the microenvironment and promote pancreatic regeneration when required by stress challenges. Future research will be needed to look at the genomic landscape of Sox9 and Sox9 mutants described in this paper in the presence of the cofactors. In addition, further experiments will be needed to validate the direct binding of the cofactors to Sox9 or if other Sox9 co-binding proteins are involved in a cellular context. The specific mechanisms by which Sox9 bound with cofactors could modulate transcription factors such as PT1 and NGN3 in both the pancreatic progenitor and aberrant carcinogenic pathways can be teased out by studying the individual mutants in a cellular context.

## 4. Materials and Methods

### 4.1. Production of SOX9

#### 4.1.1. Construct Design and Cloning

Codon-optimized, synthetic genes encoding full-length Sox9 were acquired from GenScript, USA. A pair of oligonucleotides representing forward (5- CAG TCA TAT GAA TCT GCT GGA TCC G -3) and reverse (5 -CAT ACT CGA GTC ATT TAT AGT CCG GAT G- 3) sequences, along with NdeI and XhoI restriction sites, were synthesized (Integrated DNA Technologies, Coralville, IA, USA) to amplify the Sox9 gene fragment (construct containing HMG and N-terminal domain, 1–173 amino acids) from the full-length gene using a standard polymerase chain reaction (PCR) protocol. The amplified PCR product was purified by agarose gel electrophoresis and subsequently digested with restriction enzymes NdeI and XhoI (New England Biolabs, Ipswich, MA, USA). The digested PCR product was ligated into pET-28a vector (Novagen, Madison, WI, USA), which was digested with the same set of restriction enzymes. The integration of construct with the vector was verified by DNA sequencing. Similarly, Sox9 HMG domain and other constructs were subcloned into pET-28a vector.

#### 4.1.2. Site Directed Mutagenesis

All the point mutations (A76E, H104A, K122A, K141A, H165A, K166A, and K167A) in the Sox9 construct (1–173) were introduced by site-directed mutagenesis using an overlap extension PCR method. The QuickChange II XL Site-Directed Mutagenesis Kit (Agilent Technologies, Santa Clara, CA, USA) was used and suitable oligonucleotide pairs were purchased from Integrated DNA Technologies. DNA sequencing using vector specific primers in both the forward and reverse directions confirmed the mutation at the desired position.

### 4.2. Protein Expression and Purification

For the expression of the Sox9 (1–173) construct, a single colony of *E. coli* BL21(DE3)-RIL strain carrying the desired construct was inoculated in 100 mL of Luria-Bertani medium supplemented with 50 μg.mL^−1^ kanamycin and 50 μg.mL^−1^ chloramphenicol, and grown overnight at 37 °C in a rotary shaking incubator. An aliquot (1%) of the overnight-grown seed culture was inoculated in fresh Luria-Bertani medium supplemented with the same antibiotics and allowed to grow further until the absorbance (A) at 600 nm reached a value of 0.6–0.8. The cultures were then down-tempered to 18 °C for 1 h before induction with 0.2 mM IPTG (isopropyl β-D-1-thiogalactopyranoside) for 16 h at 18 °C. The cells were harvested by centrifuging the culture at 18,671× *g* for 15 min at 4 °C. The cell pellet was resuspended in lysis buffer (50 mM Tris-HCl, pH 8.0, 500 mM NaCl, 2 mM beta-mercaptoethanol (β-ME), and 20 mM Imadazole) along with a mixture of protease inhibitors and DnaseI. The resuspended cells were disrupted by sonication (30-s pulse on/off with 40% amplitude) (Sonics, Newtown, CT, USA). The soluble and insoluble cell fractions were separated by centrifuging the cell lysate at 41,657× *g* for 60 min at 4 °C. The supernatant was loaded onto a nickel-nitrilotriacetic acid (Ni-NTA) column that was pre-equilibrated with equilibration buffer (50 mM Tris-HCl, pH 8.0, 500 mM NaCl, and 20 mM Imadazole). The column was washed with 25 column volumes of wash buffer (50 mM Tris-HCl, pH 8.0, 500 mM NaCl, 2 mM β-ME, and 40 mM Imadazole) to get rid of any nonspecifically bound proteins. Finally, the protein was eluted with elution buffer (50 mM Tris-HCl, pH 8.0, 150 mM NaCl, 2 β-ME, and 300 mM imidazole) and subsequently dialyzed overnight against 25 mM Tris-HCl, pH 8.0, 150 mM NaCl, 2 mM β-ME in a cold room. The dialyzed protein was further purified by gel filtration chromatography using a Superdex 200 PG 16/30 column (GE Healthcare, Chicago, IL, USA) equilibrated with 25 mM Tris-HCl, pH 8.0, 150 mM NaCl, 1 mM DTT. The purified protein was concentrated using an Amicon concentrator with a 10-kDa-molecular mass cutoff membrane and the concentration was measured by UV 280 nm using calculated extinction coefficients. The mutant proteins A76E, H104A, K122A, K141A, H165A, K166A, and K167A were expressed and purified similarly to the wild type. The purity of the protein at each stage was checked by 4–20% SDS-PAGE.

### 4.3. UV-Visible Spectroscopy

Purified Sox9 proteins of 2–4 mg/mL concentration were used for reading the absorbance of bind cofactors. The UV-Vis spectra were obtained on a UV-Vis spectrophotometer BMS (Biotechnology Medical Science) UV-1602, scanning from 300 nm to 800 nm. The baseline was corrected with the buffer used to prepare the protein samples.

### 4.4. Sequence Alignment

Sequences for Sox9 (mouse, human) and several Sox member proteins were obtained from UniProt (Universal Protein Resource, URL: https://www.uniprot.org/). Multiple sequence alignment was performed using Clustal Omega (https://www.ebi.ac.uk/Tools/msa/clustalo/). It uses seeded guide trees and HMM (Hidden Markov Model) profile–profile techniques to generate alignments [40]. Further annotation where secondary structures are displayed on the top of the alignment for Sox proteins for HMG part was performed by ESPript. Identical residues are shown in blue, whereas similar residues are shown in red. The figure was generated through ESPript [41].

### 4.5. Mass Spectrometry

#### 4.5.1. Sample Preparation for LC/MS/MS

Each sample in the SDS-PAGE loading buffer was loaded on a NuPAGE 4–12% BisTris gel in MOPS SDS buffer (Invitrogen, Waltham, MA, USA) and then stained with Coomassie Blue G-250 (Sigma, St. Louis, MO, USA) for 3 h. Each major gel band was fractionated and then destained by 40% methanol. For trypsin digestion, the samples were treated with 500 ng of trypsin (Promega, Madison, WI, USA) at 37 °C overnight. After digestion, the peptide samples were dried and stored at −20 °C for further analysis.

#### 4.5.2. LC/MS/MS Analysis

A tryptic digest mixture of each sample was injected and separated by an Easy nLC-1000 system (Thermo Fisher Scientific, Waltham, MA, USA) equipped with a nano-electrospray ionization source and on in-house packed column emitters using a 20 cm fused silica capillary (i.d. 75 μm, o.d. 360 μm) filled with C18 (particle size 3 μm, 100 Å) resin (Dr. Maisch GmbH, Ammerbuch, Germany). The peptides were eluted using a 120 min gradient from buffer A (0.1% formic acid in water) to buffer B (0.1% formic acid, 100% acetonitrile). Eluted peptides were ionized by 2.0 kV spray voltage and introduced into a Q Exactive Plus mass spectrometer (Thermo Fisher Scientific). The analysis method consisted of a full MS scan with a range of 400–1650 *m/z* at 70,000 resolution, data-dependent MS/MS (MS2) on the 15 most intense ions from the full MS scan, dynamic exclusion for 60s, higher energy collisional dissociation (HCD, collision energy 25), and the exclusion of unassigned charge states and singly charged ions.

#### 4.5.3. Mass Spectrometry Data Analysis

MS data were analyzed and annotated with Genedata Expressionist software (v.13.0.1) using the Peptide Mapping Tool. The raw MS data were processed using two Genedata modules: Refiner MS for data pre-processing, and Analyst for data post-processing and statistical analyses. Briefly, after noise reduction, LC-MS1 base peaks were detected and their properties were calculated (*m*/*z* and RT boundaries, *m*/*z* and RT center values, intensity). Chromatograms were further aligned based on the RT spectra using a 2 min range. Individual peaks were grouped into clusters, and MS/MS data associated to these clusters were annotated with the Peptide Mapping Tool using the FASTA sequence of sp|Q04887|SOX9_MOUSE Transcription factor SOX-9. The search was performed with a peptide tolerance of 0.1 Da, a MS/MS tolerance of 0.01 Da, two allowed missed cleavages, and only annotations with a minimum score of 10 were considered. As variable modifications, the following parameter were applied: carbamidomethyl (C) maximum: 2 per sequence allowed: anywhere; FAD (C), FAD (H), FAD (Y) maximum: 2 per sequence allowed: anywhere; FMN (S), FMN (T) maximum: 2 per sequence allowed: anywhere; pyridoxal phosphate (K) maximum: 2 per sequence allowed: anywhere; and pyridoxal phosphate H2 (K) maximum: 2 per sequence allowed: anywhere.

### 4.6. Cellular Assay

Human-induced pluripotent stem cells (hiPSCs) generated in Dr. Abdelalim’s lab at QBRI were cultured and maintained using mTeSR Plus medium (Stem Cell Technologies, Vancouver, BC, Canada) and then plated on 1:50 Matrigel-coated dishes (Corning, New York, NY, USA). Upon reaching 70–80% confluency, cells were differentiated into pancreatic progenitors using Memon et al.’s protocol [42]. Cells were treated with different concentrations of FAD (0.25 µM, 0.5 µM, and 1 µM) (Sigma-Aldrich, Darmstadt, Germany) or PLP (4 µM and 8 µM) (Sigma-Aldrich, Germany) during the four days of pancreatic progenitor stage of in vitro differentiation. The control cells were treated with water.

### 4.7. Immunostaining and Western blotting

After the treatment period, cells were fixed, and immunostaining was performed using previously reported protocols [42]. Cells were also collected for total protein extraction using RIPA lysis buffer and were normalized to 20 ng/µL before SDS-PAGE separation. Nitrocellulose membranes were blocked using 15% skimmed milk in 0.5% TBST. Membranes were then incubated with primary antibodies overnight at 4 °C. Membranes were washed with 0.5% TBST and secondary antibodies were added for 1 h at room temperature. Membranes were developed using SuperSignal West Pico Chemiluminescent substrate (Pierce, Loughborough, UK) and were then visualized using an iBright™ CL 1000 Imaging System (Invitrogen).

### 4.8. RT-qPCR and Statiscial Analysis

RNA was extracted using Direct-zol™ RNA Miniprep (Zymo Research, USA) following the manufacturer’s protocol. cDNA was then synthesized from 1µg of RNA using a SuperScript™ IV First-Strand Synthesis System following the manufacturer’s protocol (ThermoFisher Scientific, Waltham, MA, USA). RT-qPCR was performed using GoTaq qPCR Master Mix (Promega, Madison, WI, USA), a SYBR Green-based detection system, and amplification was detected using a Quant Studio 7 Flex system (Applied Biosystems, Waltham, MA, USA) run in triplicates. Average Ct values were normalized to the control/WT samples for each gene tested. GAPDH was used as an endogenous control. Analysis was performed using Microsoft Excel and the relative RNA expression was calculated using ΔΔCT. Statistical analysis was performed using an unpaired two-tailed Student’s *t*-test by Prism 8 software. NGN3: 5′- GGCTGTGGGTGCTAAGGGTAAG -3′, 5′- CAGGGAGAAGCAGAAGGAACAA -3′; PTF1A: 5′- ATTATGGCCTCCCTCCCCTA -3′, 5′- AGTTTTCTGGGGTCCTCTGG -3′; GAPDH: 5′- ACGACCACTTTGTCAAGCTCATTTC -3′; 5′- GCAGTGAGGGTCTCTCTCTTCCTCT -3′.

### 4.9. Structural Analysis

Structures of mouse (4S2Q) and human (4EUW) Sox9 available in the RCSB Protein Data Bank (PDB, URL: https://www.rcsb.org) were analyzed using the RCSB online viewer, as well as UCSF Chimera; 3D coordinates were superimposed and figures were produced using UCSF Chimera [43]. 

### 4.10. Disease Association of SOX9 Residues

The disease association of the following residues, A76, K122, K141, K166, K167, H104, and H165 was investigated using variant information in UniProt (Universal Protein Resource, URL: https://www.uniprot.org/), and somatic mutation information in COSMIC (Catalog of Somatic Mutations in Cancer URL: https://cancer.sanger.ac.uk/cosmic/).

## Figures and Tables

**Figure 1 ijms-23-14051-f001:**
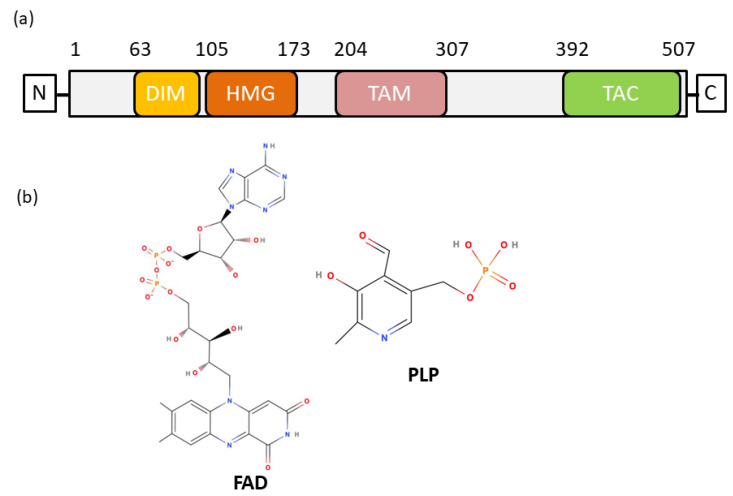
**Structural and functional domains of the SOX9 and chemical structure of FAD and PLP**. (**a**) Domain organization of the Sox9 protein. Dimerization domain (DIM) is located at N-terminus and precedes the HMG domain. The HMG domain is followed by two separate transactivation domains, TAM and TAC. (**b**) Chemical structure of FAD and PLP.

**Figure 2 ijms-23-14051-f002:**
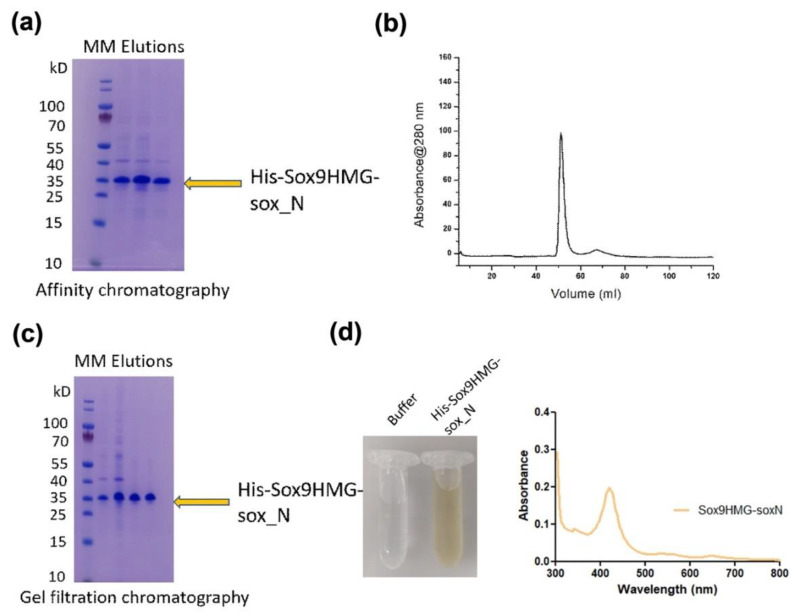
**Purification and Uv-Vis spectroscopy of Sox9 construct**. (**a**) Sox9HMG-sox_N construct (amino acids 1–173) was expressed in bacteria and purified using Ni-IMAC affinity chromatography, owing to having 6xHis at N-terminal of the construct. (**b**) Protein was concentrated and subsequently purified using gel filtration chromatography. The gel filtration chromatogram showed a homogenous and monodisperse peak. (**c**) SDS-PAGE of the gel filtration elution fractions corresponding to the Sox9HMG-sox_N construct. (**d**) Uv-Vis spectroscopic spectra of the purified Sox9HMG-sox_N showing absorption maxima between 400–450 nm. Buffer was used as a reference.

**Figure 3 ijms-23-14051-f003:**
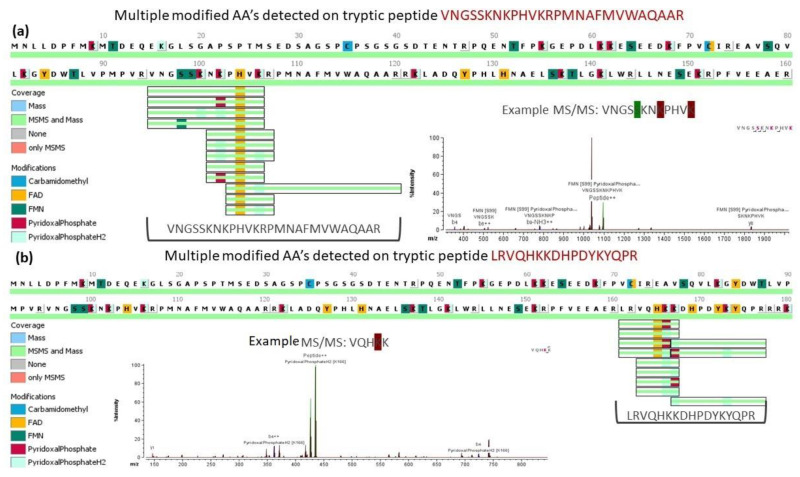
**MS/MS spectra showing modifications:** MS spectra shown for representative peptide showing the highlighted residues where both mass and MS/MS data were obtained with example for FAD (**a**) and PLP (**b**).

**Figure 4 ijms-23-14051-f004:**
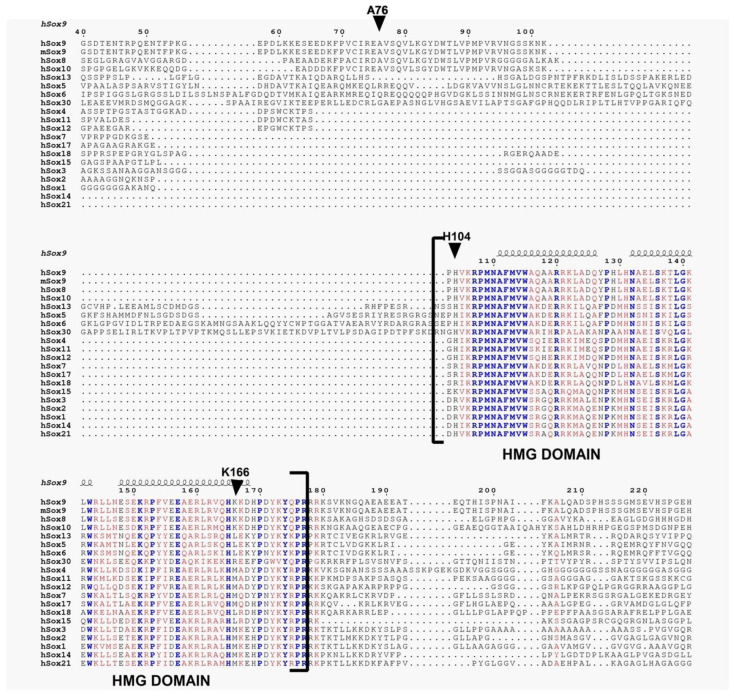
**Multiple sequence alignment of Sox family of proteins.** The secondary structures are displayed on the top of the alignment for HMG domain. Identical residues are shown in blue, whereas similar residues are shown in red. We have omitted initial N- terminal and the C-terminal alignment portion from the final figure.

**Figure 5 ijms-23-14051-f005:**
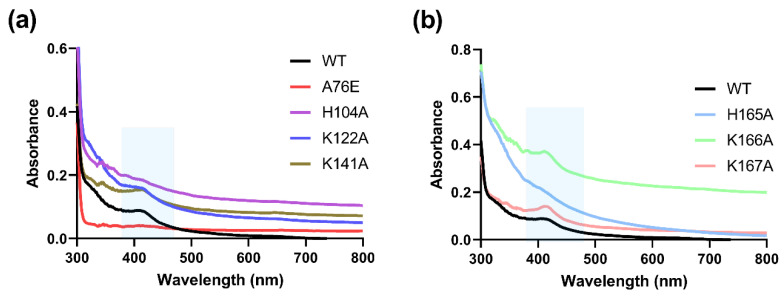
**Absorbance spectroscopy of Sox9 and mutants.** Absorbance of Sox9HMG-sox_N domain and key site directed mutants showing the loss of peak at ~420 nm for (**a**) A76E and H104A and (**b**) H165A.

**Figure 6 ijms-23-14051-f006:**
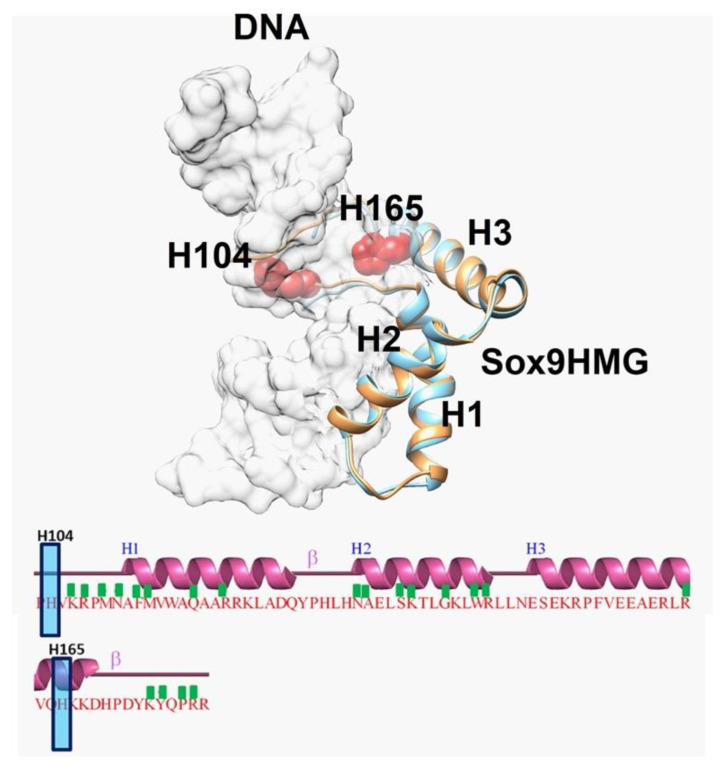
**Superposition of Sox9 structures showing key residues implicated in dimerization:** Superposition of mouse Sox9HMG domain structure (PDBID: 4S2Q) in complex with DNA with human Sox9HMG domain structure in complex with DNA (PDBID: 4EUW). mSox9HMG is in brown, whereas hSox9HMG is in light blue. For clarity purposes, DNA is shown as light grey in transparent surface representation. The HMG domain contains three helices (H1, H2, and H3), and their interactions with DNA are represented with a green box. H104 and H165 are not part of the interaction and are shown as red.

**Figure 7 ijms-23-14051-f007:**
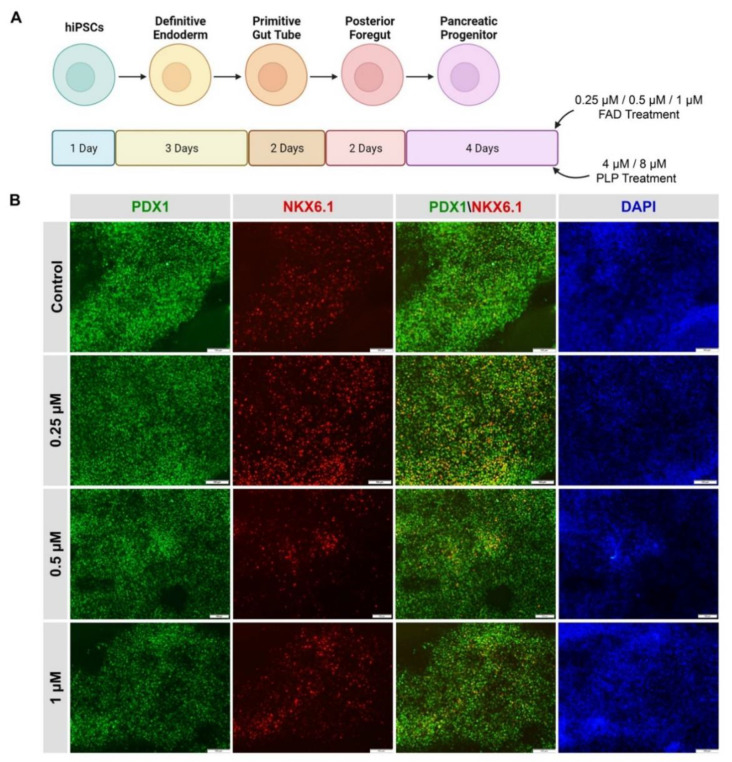
**Differentiation of hiPSCs into pancreatic progenitors in the presence of FAD and PLP.** (**A**) A diagram showing the steps of pancreatic progenitor differentiation and their treatment with FAD or PLP. During stage 4 of differentiation, the cells were treated for 4 days with different concentrations of FAD (0.25 µM, 0.5 µM, and 1.0 µM) or PLP (4 µM and 8 µM) (**B**) Representative immunostaining images showing the co-expression of PDX1 and NKX6.1 in iPSC-derived pancreatic progenitors after treatment for 4 days with different concentrations of FAD. Scale bars = 100 µm.

**Figure 8 ijms-23-14051-f008:**
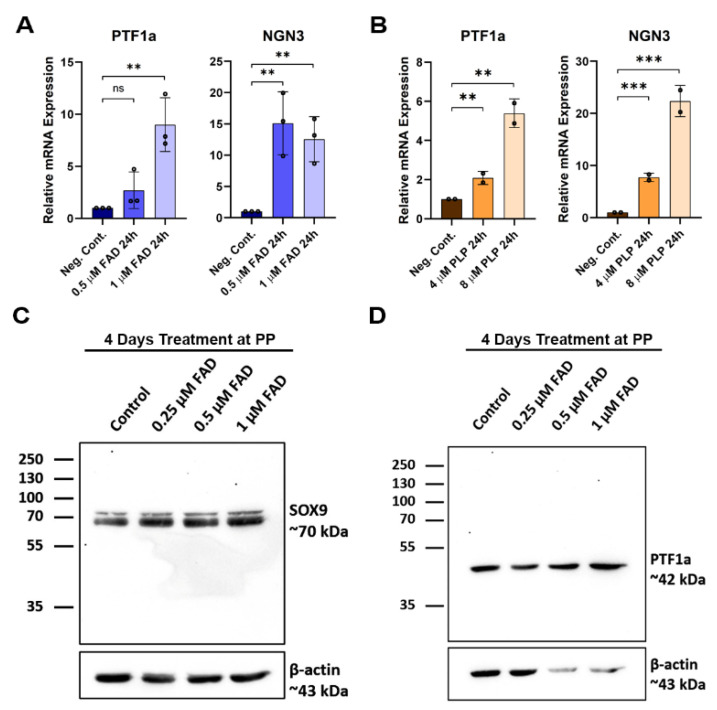
**The effect of FAD and PLP treatment on the expression of key pancreatic progenitor transcription factors.** RT-qPCR analysis of the PTF1a and NGN3 expression after treating iPSC-derived pancreatic progenitors for 24 h with FAD (**A**) or PLP (**B**). Western blot analysis for the expression of SOX9 (**C**) and PTF1a (**D**) after treating iPSC-derived pancreatic progenitors for 4 days with FAD. Data are represented as mean ± SD; ** *p* < 0.01, *** *p* < 0.001.

**Figure 9 ijms-23-14051-f009:**
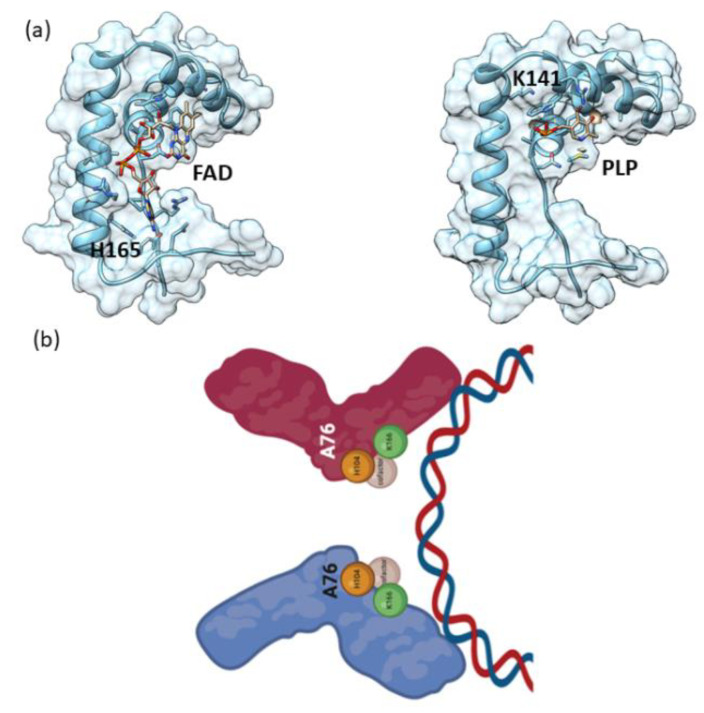
In silico/docking Sox9 HMG-FAD/PLP interaction. (**a**) Cartoon as well as surface representation of the Sox9 HMG bound to FAD/PLP. Protein was rendered according to secondary structure elements. The complex structure highlighting the binding interfaces. (**b**) Schematic diagram of the cofactor binding to specific residues with the dimer formation leading to binding to DNA.

## Data Availability

Not applicable.

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
