# Peer review of "Flavin Adenine Dinucleotide (FAD) and Pyridoxal 5′-Phosphate (PLP) Bind to Sox9 and Alter the Expression of Key Pancreatic Progenitor Transcription Factors"

_ijms, 2022, doi:10.3390/ijms232214051_

Round 1
Reviewer 1 Report
Islam et al., reports the role of small molecule ligands FAD and PLP in the expression of key pancreatic progenitor transcription factors through the interactions with Sox9, a key transcription factor in early pancreatic development, using a combination of data from mass spectrometry, UV absorption spectrometry, and mutagenesis studies. The findings of this paper are very interesting and deserve to be published in IJMS, however, prior to the acceptance, the authors must consider the following points for revising this manuscript.
Specific points
1. Provide a bar representation diagram for Sox9 to illustrate the architecture, and label the key a.a and regions.
2. Pg 3: Materials and Methods - for the centrifugation only the rpm is provided. The rotor info must be given along with the rmp, if not, provide the “g” value.
3. Figure 1. Panel b and d include the standard MW elution profile, and label all the peaks. If possible, please provide a better or improved gel for panel c the right-hand side.
4. Pg 6: line 232 and elsewhere provide space between the number and unit.
5. Pg 8: line 272: the rmsd of ~ 1Å for how many Ca atoms? Indicate the number of atoms superimposed.
6. Authors wish to provide an in silico complex model for the binding of ligands FAD and PLP with Sox9. Label the key residues that were mutated and verified their role. In the absence of in vitro binding affinity, the inferred affinity numbers using in silico approach might be provided. Moreover, the chemical structures of FAD and PLP should be given.
7. As a part of the discussion section, authors should consider providing a schematic diagram that captures the whole message of this paper. This will help the readers to understand the findings of this paper.
Author Response
Please see the attachment. I have attached file with comments/responses to both reviewers.

Reviewer 2 Report
Comments on Manuscript ID: ijms-2007661, entitled "Flavin adenine dinucleotide (FAD) and pyridoxal 5’-phosphate (PLP) bind to Sox9 and alter the expression of key pancreatic progenitor transcription factors" by Islam et. al.
The paper describes the FAD and PLP binding to SOX9 protein and their role in pancreatic progenitor transcription factors to alter their expression using different structural studies to validate the binding mechanism.
Following things need to be explained in the MS.
1. Figure 1-5: Purified His-SOX9-HMG-Sox-N proteins as well as different mutants of the SOX9 protein with role of different amino-acids in HMG domain to FAD and PLP binding are nice to study the binding mechanism. Figure 6-7 shows that FAD increase leads to increase in PDX1 and decrease in NKX61 expression of mutant SOX-9 proteins and PTF1a, NGN3 and SOX-9 protein using Western blotting. The author has not shown the role of SOX9 mutant proteins that how those affect the differentiation or programming of these progenitor cells w.r.t. these transcription factors, especially the one found to be clinically relevant with colon adenocarcinoma as author reported on line 255?
2. The role of these mutants SOX9 proteins in cellular models in context to pancreatic beta cell programming for glucose tolerance/diabetic model would be nice to study and evaluate whether FAD and PLP act as controlling metabolites to regulate the SOX9 expression.
3. Figure 6-legend needs more detailed explanation of the figure.
4. Needs proofreading by all authors-font size variation-e.g. line 381-389.
The paper is technically sound and well written. Comments above- will make the MS better.
Author Response
Please see the attachment. I have attached comments and responses to both reviewers here.
